# Sustainable Economic Solution to Prevent the Wasted Cold Water in the Start of Shower Time Using Low Cost In-Line Electrical Heater with MISO Fuzzy Logic Controller

Muhammad M. A. S. Mahmoud 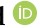

Automation Process Engineering Department, Baku Higher Oil School, Baku AZ1025, Azerbaijan; mmanar@yahoo.com

**Abstract:** This paper discusses domestic problem of waiting hot water for the shower use till it reaches satisfactory temperature, which result a lot of wastage in fresh water. The outcome from research survey shows that there is no satisfactory solution till now as all solutions were either expensive or with no effective results. Local small inline electric heater equipped with fuzzy logic controller is proposed in this paper to be installed just before the showerhead to measure the water temperature and flow before the showerhead, as control input-variables, and decide the operating voltage of the heater, as control output-variable. Matlab Simulink is used to model the proposed system. Different test cases are simulated to prove the performance and the safe operation of the system. Techno-economic study is carried out to determine the "Direct Benefits" and "Indirect Benefits" that can be achieved if such system is implemented in wide range. Azerbaijan data is taken as an example to calculate the economic benefits. The results show important benefits not only for economy, that can increase fresh water sales opportunity to 545,088,171.52 $, but also for climate and the reduction of greenhouse gas emission. Different economic indices are provided to be an easy reference for decision makers and project managers.

**Keywords:** Azerbaijan sustainable development; water electric heating; energy efficiency; green buildings; fuzzy logic temperature control; greenhouse gas emission

## 1. Introduction

Freshwater scarcity and its crisis is a major concern in the 21st century [1,2]. This issue has created a lot of serious problems worldwide between countries that share same water resources such as the conflict between Nile river basin countries started on 2015, even earlier. Therefore, looking to the future, world need to seriously consider rationalize the water consumption, and hence reduce the water wastage as much as possible [3–5].

Wastewater is a term for water with its quality affected via 'unwanted' anthropogenic sources thus the word 'waste'. Types of wastewater can include those from domestic household, commercial, agricultural etc. Domestic wastewater or better known as municipal wastewater is conveyed to sewers. Municipal sewage consists mostly of greywater (showers, dishwasher sinks, cloths-washer etc.), black-water (toilet flushes with human waste an excreta), soaps and detergents. Processing this wastewater is expensive [6].

Most of us waste a lot of water waiting for hot water to reach the sink or showerhead. In a large house, piping length of fifty feet, even more, is used to connect the water heater to the most distant tap. And with 50 feet of pipe, it can take a long time for hot water to reach the tap, and while waiting, all the water goes down the drain.

Waiting time for hot water depends on three factors: the distance from the water heater, the diameter of the piping, and the flow rate. The effect of distance is pretty obvious. If longer distance the hot water has to reach, longer time it will take to get there. In a new house, you can design short plumbing by locating bathrooms and the kitchen near each other.

The effects of pipe diameter and flow rate are not so intuitive. The smaller the diameter, the faster hot water will reach the tap. That's because smaller-diameter pipe holds less water. Fifty feet of 1/2"-diameter pipe holds 0.8 gallons, while the same length of 3/4" pipe holds 1.4 gallons and 1" pipe holds 2.3 gallons. When the user is waiting for hot water, all of the cooled-off water sitting in the pipe has to flow out before hot water from the water heater reaches the tap.

The flow-rate of a faucet or showerhead affects the wait for hot water because it governs how quickly the cooled water sitting in the pipes will be emptied out. If the user has a water-conserving bathroom faucet that delivers only a half-gallon per minute (gpm) and most of the piping in the house has 3/4" piping, a 50-foot run from the water heater requires almost three minutes to get hot water to your sink. The irony is that even as the flow rates of faucets and showerheads has dropped, plumbing codes are increasingly mandating larger-diameter piping, so the wait times for hot water have increased. A fact that's exacerbated by the larger houses are being built.

In addition to the long wait for hot water, all of the cooled-off water sitting in the pipe goes down the drain. Counting nationally the amount of water wasted while waiting hot water, there's a huge amount of water is wasted in this way. Additional water waste will occur if the shower is turn on tend on then the person left to do something else until hot water is received. This all-too-common practice can be extremely wasteful, especially if person leaves the bathroom and get distracted after turning on the water.

In domestic application, different approaches are followed to solve this problem of waiting hot water and the consequent wasting of cold water [7–9]. The main approaches that widely used are:

### 1.1. Circulatory System

Circulatory systems are commonly used in the design of warm water systems to reduce the time taken for the heated water to reach the outlet based on the following equation:

$$t = \frac{C}{F} \tag{1}$$

where:

t = Time delay—Amount of time for heated water to reach the outlet (m),
C = Capacity (litter),
F: Flow rate of tap (litters/min).

Less waiting time is achieved by circulating heated water through the system and returning cold water to the water heater using a pump or pumps set, usually installed on the heater return line. Figures 1 and 2 illustrate two different circulatory systems.

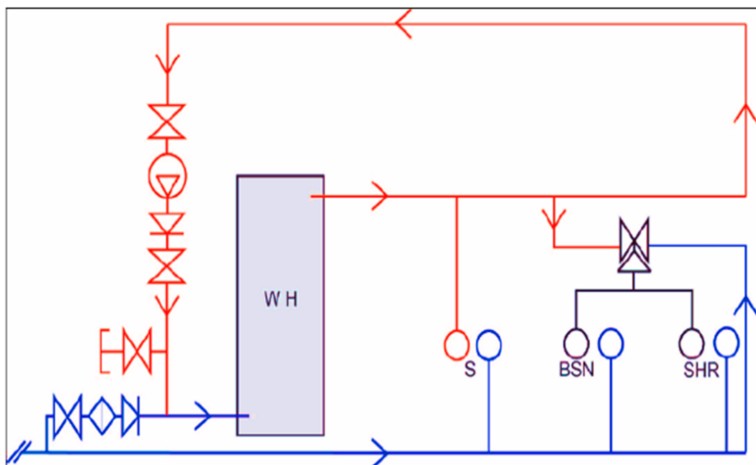

**Figure 1.** Example of a circulating warm water system controlled by a thermostatic mixing valve.

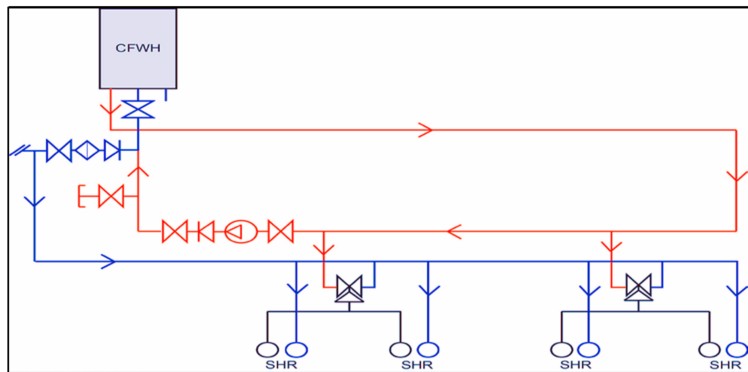

**Figure 2.** Example of a circulating warm water system controlled by a continuous flow water heater.

Where:

- S: Sink
- SHR: Shower
- BSN: Basin
- WH: Water Heater
- CFWH: Continuous flow water heater
- Blue line: Cold Water pipeline
- Red line: Warm Water pipeline

Correct pump selection is critical as they must have sufficient design pressure and flow rate for the system. Incorrect pump selection can cause excessive energy consump-tion and an under-sized pump may result in hot water plumbing system operating at a much higher and fluctuating temperature whereas an oversized pump may result in ex-cessive water velocities. Also, operating such pump continuously, will consume a lot of electrical energy that will be added to the water supply invoice.

### 1.2. On-Demand Circulator

Another system is "on-demand circulator". With this system, a user in a remote bathroom or kitchen pushes a button to activate a small pump, usually under the sink. This pump pulls hot water from the water heater, and sends the cold water that's been sitting in the pipes back to the water heater, either through a separate piping return that has been installed (most common in new construction), or via the cold water line (more common in retrofit applications). As soon as hot water reaches the tap, a temperature-controlled switch turns off the pump (Figure 3). The system is also available with an occupancy sensor to automatically turn on the circulator pump when someone enters the bathroom, though this will lead to some energy wastage since the circulator will operate whether or not hot water will be needed. This system cost almost 300 USD.

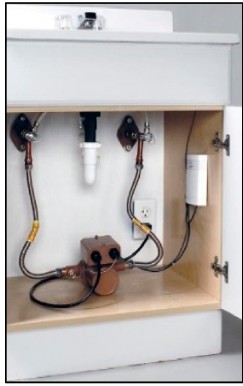 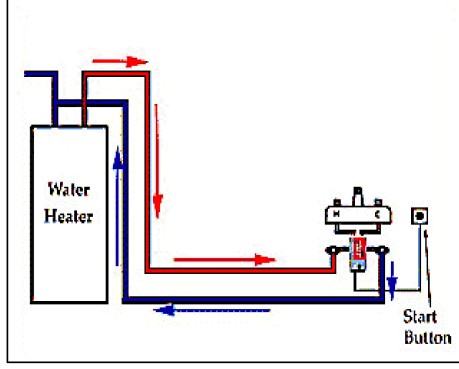

**Figure 3.** On–demand circulator.

Where:

- CFWH: Continuous flow water heater,
- Blue line: Cold Water pipeline,
- Red line: Warm Water.

This method is effective to reduce the waiting time for hot water, but still it need time to pull the hot water from the central heater till the service point. This duration depends mainly on the distance between the service point and the center heater, also on the diameter of the hot water pipe. Another limitation need to be considered, that the central heater need to be suitable to operate under negative pressure. Also, this system is quite expensive and its cost may even more than the price of some central heaters, in additional to its electrical consumption cost.

Finally, systems such as Circulatory systems and On-demand circulator consist of rotating equipment (pump and motor), and in general rotating equipment have less reliability rates than static equipment (the central heater), hence the availability rate of the entire hot water system reduces by installing these systems [10].

### 1.3. Distributed Instant Electrical Water heaters

These provide instant hot water to a single appliance and are usually electrical heaters. There are two basic types:

#### 1.3.1. Hand Wash Type

This is small water heaters, usually around 3 kW output, that are fitted over a single washbasin. They are inlet controlled. The temperature of the water depends on the flowrate through the heater. The faster the flowrate, the cooler the water. Flow rates are generally poor but adequate for hand washing.

#### 1.3.2. Electric Shower Type

The outputs of these heaters are up to 12 kW, many electric showers feature sophisticated microchip -technology allowing the stabilization of the temperature at low and high flow rates. Electric showers feature a low-pressure cut-out to guard against scalding if the pressure or flow rate suddenly drop. Flow of water is controlled by an electrically operated solenoid valve that operates when the electricity supply is "on" to the heater.

This system is smart hot water supply system, and it can provide instant hot water at service point. But the disadvantage of such equipment is it high initial and running cost. Also, this equipment introduces higher greenhouse gas emissions if not powered by a renewable energy source.

### 1.4. Polyethylene (PEX) Piping Systems

Using cross-linked polyethylene (PEX) piping systems as thermal insulated piping system instead of galvanized piping system gives opportunity to reduce water and energy waste by reducing the amount of time to deliver hot water to the outlet from the water heater. Also, selection of the installation type between Trunk & Branch, Home Run and Remote Manifold affects the wait time for hot water.

Tests were also performed on each of the three PEX system designs to compare the time it takes for hot water to be delivered to the test fixture (TF). Figure 4 shows the results of delivering hot water to the showerhead after the cold water were flushed out from the hot water pipe. The results were normalized to keep the flow rates and temperature from the hot water tank constant for all systems.

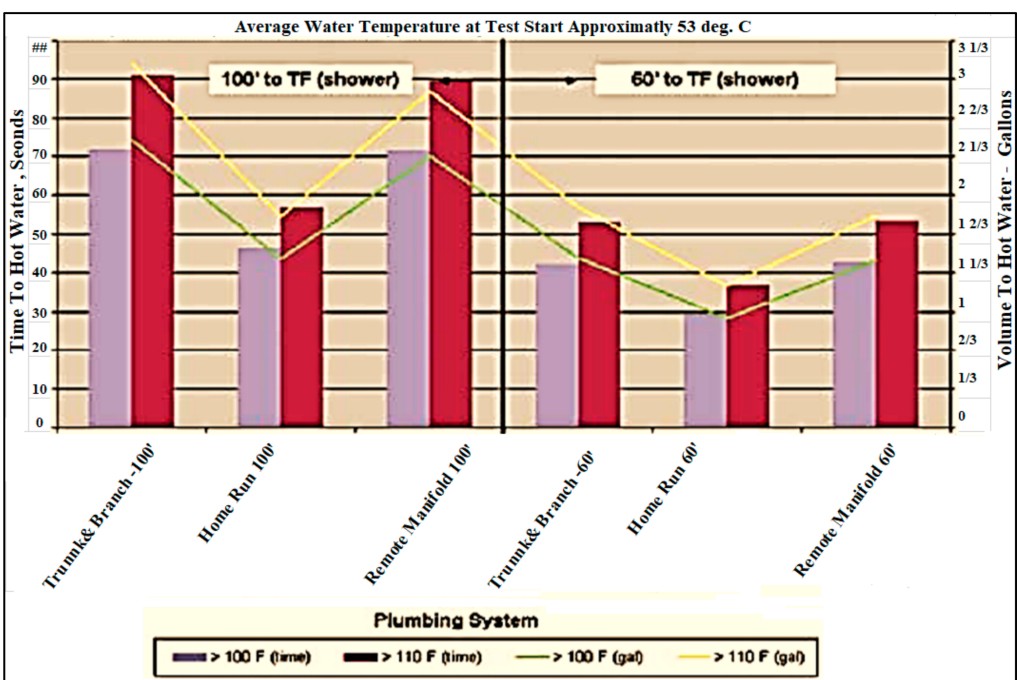

**Figure 4.** Comparison of Hot Water Delivery Time.

Water and time savings between 30% and 40% were identified based on this analysis of the home-run system over either the trunk and branch or remote manifold system designs. The data indicates:

- Home-run systems will deliver hot water to the outlet quicker, especially when the pipes are at room temperature
- Trunk and branch and remote manifold systems will deliver hot water quicker during sequential flows.

In [11], project seeks to quantify the magnitude of hot water waste in existing Northeastern homes in USA was executed. The monitoring effort focuses on measuring the amount of hot water waste in five homes. The five monitored test sites include are grouped as following:

- Group 1—two baseline (or control) sites
- Group 2—one site where a conventional hot water tank was changed to a tankless system
- Group 3—two sites where the water heater and distribution system are changed.

The results showed that there were some indications of improved hot water "usefulness" at the fixture for one of the two retrofitted sites. However, no direct reduction in hot water use or energy use was measured. Therefore, no energy cost savings were observed to offset the installed cost of $500–$1000 to implement these distribution system improvements.

For further analysis of the results given in [11], although tankless water heater system is used in Group 2, still a startup period of 40-s is required to receive hot water with acceptable temperature, with no energy saving. At Group 3, site 1 hot water supply line was shortened by 177 in. and hot water recirculation pump was installed, but still the result was not satisfactory, and no recognized benefit in saving energy or water.

This paper considers both existing installations and new installations in providing cost-effective smart solution to this domestic problem of waiting hot water for the shower use, which result wasting a lot of fresh water. In this solution, a new concept of using in-line electrical heater with fuzzy logic controller is introduced. In Section 2, design consideration for the proposed water heater system is discussed. Modelling of Central gas heater and instant electrical heater using Simulink is introduced in Section 3. Fuzzy controller design is provided in Section 4. Testing and Commissioning the model is given

in Section 5. Techno-Economic analysis for the proposed solution is given in Section 6. Finally, conclusion is given in Section 7.

## 2. Design Consideration

In this section, the design criteria and limitation such as safe operating temperature, flow, utility water supply temperature is discussed.

### 2.1. Water Supply from Utility

Minimum water temperature and pressure received at service point from utility is considered to be 10 °C and 2 bar. These values are used as input parameters to the hot water heating system.

### 2.2. Safety Temperature

In general, safe warm water is specified in to be approximately between 38 °C and 48 °C [12]. The comfortable warm water for shower purpose should be between 40.5 °C and 43.5 °C [8]. In this paper, approximately temperature of 44 °C is considered in the design of the inline heater controller.

### 2.3. Minimum and Maximum Flow

To assign the maximum hot water flow during shower, manufacturers' standard recommends 2 gpm [13], This flow is in line with the general comfortable rate for shower flow and ASME standard [14]. To assign minimum flow rate, safety precaution is takin in order not to allow the temperature of the water flows inside the in-line electrical heater to reach close to water boiling point when the electrical heater is producing maximum temperature. This minimum flow rate value is taken as 0.87 gpm approximately. In Section 3, supporting equations is provided.

### 2.4. Main Hot Water System

The main heater that serve the entire domestic hot water system can be any type such as electrical or gas water heater with or without tank, with or without central recirculation pump. The piping for the domestic hot water system can be any non–metal or metal, insulated or not insulate, imbedded inside the wall or surface mounted.

## 3. Modelling

In this section, the concept and the main component modeling of the proposed system is discussed. Figure 5 illustrates the block diagram for the proposed system to solve the problem of waiting hot water for shower use.

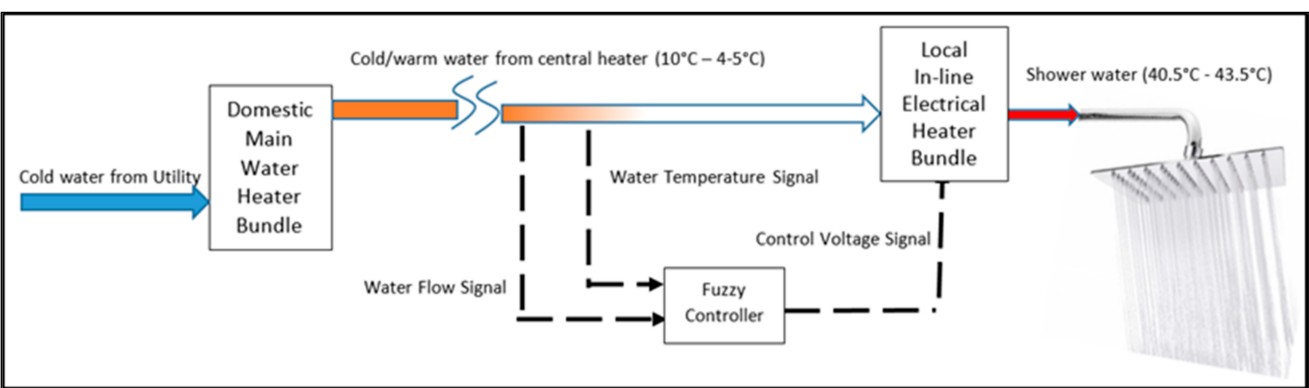

**Figure 5.** Block diagram for the proposed system.

### 3.1. Design Concept

The design concept is based on the idea of installation of small inline automatic electrical water heater just before the showerhead to heat up the cold water instantaneously to 40.5–43.5 °C until the warm water is received from the main water-heater, then automatically the inline electrical heater is disconnected. Fuzzy logic controller is used to control the inline electrical heater. The controller has two signals; first signal is water temperature measured just before the showerhead, and the second signal is the water flow going to the showerhead. Utility power can be used to energies this small inline heater, but, for more economic operation, electrical energy from solar cell can also be used.

### 3.2. Electrical Heater Element Modeling:

The model of electrical heater consists of thermal electrical resistance to generate the required heat. This heat energy is transferred to the water using heat exchange bundle [15].

The thermal resistor model represents a temperature-dependent resistor. The following equations describes the dynamic thermal behavior of the temperature-dependent resistor model:

$$Q = K_d \, t_c \frac{dT}{dt} - IR \tag{2}$$

$$R = R_0 * (1 + \alpha_t(T - T_0)) \tag{3}$$

where:

R: Temperature-dependent resistor
Q: Net heat flow.
$\alpha_t$: Temperature coefficient.
$K_d$: Dissipation factor parameter value.
$t_c$: Thermal time constant parameter value.
dT/dt: Rate of change of the temperature.
I Current through the resistor.
$R_0$: the nominal resistance at the reference temperature $T_0$.
A: Resistance material temperature coefficient.

### 3.3. Heater Bundle

The bundle model represents a pipeline segment with a fixed volume of liquid. The liquid experiences pressure losses and heating due to viscous friction and conductive heat transfer through the pipe wall. Viscous friction follows from the Darcy-Weisbach law, while the heat exchange coefficient follows from Nusselt number correlations. Heat transfer can occur in an unsteady manner [15].

The mass conservation equation for the bundle is:

$$m_{inlet} + m_{outlet} = v * \rho \left( \frac{1}{\beta} \frac{dp}{dt} + \alpha \frac{dT}{dt} \right) \tag{4}$$

where:

$m_{inlet}$: inlet mass flow rates to bundle.
$m_{outlet}$: outlet mass flow rates from bundle.
v: pipe fluid volume.
$\rho$ is the thermal liquid density in the bundle.
$\beta$ isothermal bulk modulus in the bundle.
$\alpha$ is the isobaric thermal expansion coefficient in the bundle.
p is the thermal liquid pressure in the bundle.
T is the thermal liquid temperature in the bundle.

The momentum conservation equation for the half bundle adjacent to inlet and outlet respectively point is:

$$A(p_{inlet} - p) + Fv_{inlet} = \frac{L}{2} m_{inlet} \tag{5}$$

$$A(p_{outlet} - p) + Fv_{outlet} = \frac{L}{2} m_{outlet} \tag{6}$$

where:

A is the pipe cross-sectional area.

$p$, $p_{inlet}$ and $p_{outlet}$ are the liquid pressures in the bundle at its inlet and outlet respectively.

$Fv_{inlet}$ and $Fv_{outlet}$ are the viscous dissipation forces between the bundle volume center and bundle inlet and outlet respectively.

L: Length of the bundle.

D: Hydraulic diameter of the bundle.

The viscous friction force for the half bundle adjacent to inlet and outlet respectively point is:

$$Fv_{inlet} = -\frac{\lambda N(L + L_{Eq})}{2} \frac{m_{inlet}}{2D^2} \text{ if } Re_{inlet} < Re_1 \tag{7}$$

$$Fv_{inlet} = -\frac{f_{inlet}(L + L_{Eq})}{2} \frac{m_{inlet} |m_{inlet}|}{2\rho D_{inlet}} \text{ if } Re_{inlet} \geq Re_t \tag{8}$$

$$Fv_{outlet} = -\frac{\lambda N(L + L_{Eq})}{2} \frac{m_{outlet}}{2D^2} Re_{outlet} < Re_1 \tag{9}$$

$$Fv_{outlet} = -\frac{f_{outlet}(L + L_{Eq})}{2} \frac{m_{outlet} |m_{outlet}|}{2\rho D_{outlet}} \text{ if } Re_{outlet} \geq Re_t \tag{10}$$

where:

$\lambda$: Bundle shape factor.

N: Kinematic viscosity of the thermal liquid in the bundle.

$L_{Eq}$: Aggregate equivalent length of the local bundle resistances.

D: Hydraulic diameter of the bundle.

$f_{inlet}$ and $f_{outlet}$ are the Darcy friction factors in the bundle halves adjacent to inlets and outlet respectively.

$Re_{inlet}$ and $Re_{outlet}$ are the Reynolds numbers at bundle inlets and outlet respectively.

$Re_1$: Reynolds number above which the flow transitions to turbulent.

$Re_t$ is the Reynolds number below which the flow transitions to laminar.

The Darcy friction factors follow from the Haaland approximation for the turbulent regime:

$$f = \frac{1}{(-1.8 * \log_{10}(\frac{0.9}{Re} + (\frac{r}{3.7D})^{1.11}))\hat{ }2} \tag{11}$$

where:

f: Darcy friction factor.

r Bundle surface roughness.

The energy conservation equation for the bundle is:

$$V * \frac{d(\rho u)}{dt} = \varnothing_{inlet} + \varnothing_{outlet} + Q_H \tag{12}$$

where:

$\varnothing_{inlet}$ and $\varnothing_{outlet}$ are the total energy flow rates into the bundle through inlet and outlet respectively.

$Q_H$ is the heat flow rate into the bundle through the its wall.

The heat flow rate between the thermal liquid and the bundle wall is:

$$Q_H = Q_{conv} + \frac{k_I A_H}{D}(T_H - T_I) \tag{13}$$

where:

$Q_H$: Net heat flow rate.

$Q_{conv}$: Portion of the heat flow rate attributed to convection at nonzero flow rates.

$k_I$: Thermal conductivity of the thermal liquid at the internal fluid volume of the bundle.

$A_H$: Surface area of the bundle wall, the product of the bundle perimeter and length.

$T_H$ and TI are the temperatures at the bundle wall and at the internal fluid volume of the bundle.

Assuming an exponential temperature distribution along the bundle, the convective heat transfer is:

$$Q_{conv} = |m_{avg}|c_{Pavg}(T_H - T_{in})\left(1 - \exp\left(-\frac{hA_H}{|m_{avg}|c_{Pavg}}\right)\right) \tag{14}$$

where:

$m_{avg} = (m_{inlet} - m_{outlet})/2$ is the average mass flow rate from inlet and outlet

$C_{Pavg}$: Specific heat evaluated at the average temperature.

$T_{in}$ is the inlet temperature depending on flow direction.

The heat transfer coefficient, $h_{coeff}$, depends on the Nusselt number:

$$h = Nu\frac{k_{avg}}{D} \tag{15}$$

where

$k_{avg}$: thermal conductivity evaluated at the average temperature.

$N_u$: The Nusselt number depends on the flow regime.

The Nusselt number in the laminar flow regime is constant and equal to the Nusselt number for laminar flow heat transfer parameter value. The Nusselt number in the turbulent flow regime is computed from the Gnielinski correlation:

$$Nu_{tur} = \frac{f_{avg}}{8}\frac{(Re_{avg} - 1000)\,Pr_{avg}}{1 + 12.7\sqrt{\frac{f_{avg}}{8}} * (Pr_{avg}^{2/3} - 1)} \tag{16}$$

where $f_{avg}$ and $Pr_{avg}$ are the Darcy friction factor and Prandtl number evaluated at the average temperature. The average Reynolds number is computed as:

$$Re_{avg} = |m_{avg}|\frac{D}{A\mu_{avg}} \tag{17}$$

where $\mu_{avg}$ is the dynamic viscosity evaluated at the average temperature.

## 4. Fuzzy Controller

The function of the fuzzy controller for the inline electrical heater that installed in suitable location before the showerhead is to measure the water temperature and flow before the showerhead, as control input-variables, and decide the operating voltage of the heater, as control output-variable. The fuzzy rules are designed to adjust the water temperature received by the showerhead to be within 40.5–43.5 °C as soon as the water is started to be used. By this method it is possible to eliminate the hot water waiting time and avoided wasting water.

### 4.1. Fuzzy Input and Output Membership Function:

Temperature input-variable: The range of the fuzzy membership function for temperature input is selected to be (10–44 °C). This range of input-temperature degrees covers the expected temperature signal values for the shower water that can be received at the showerhead from the moment of after opening the water tap until the moment of receiving the hot water from the central heater.

Flow rate input-variable: The range of the fuzzy membership function for flow rate input is adjusted to be between minimum 0.87 gpm and maximum 2 gpm as discussed in Section 2.

Control voltage output variable: The selected inline heater is standard heater, single phase, 220 VAC, 3 kW rating. Accordingly, the output membership function range is selected to be between minimum 0 Volt and maximum 220 Volt.

Description of Input and output fuzzy membership functions:

Nine symmetrical fuzzy triangular membership functions are used for input and output variables. The number of the fuzzy membership is increased to nine in order to achieve higher precision in reading the input and output variables, and obtain higher accuracy from the controller.

The triangular membership defined by its lower limit "a", and its upper limit "b", and the modal value "m", so that a < m < b. In all memberships used in the inline fuzzy controller, b-m is taken equal to m-a (equal margine).

### 4.2. Fuzzy Rules

Simulink model is used to run different simulation cases that can represent adequate operation scenarios for the proposed hot water system, and hence, linguistic rules are generated to insure that the control voltage signals have proper values to tune the inline heater in order to maintain the water temperature at the showerhead always within the design temperature value (40.5–43.5 °C). Table 1 lists the linguistic rues for the proposed fuzzy controller:

**Table 1.** Fuzzy rules for the inline water heater controller.

| | | Flow Rate (F) | | | | | | | | |
|---|---|---|---|---|---|---|---|---|---|---|
| | | **M1(F)** | **M2(F)** | **M3(F)** | **M4(F)** | **M5(F)** | **M6(F)** | **M7(F)** | **M8(F)** | **M9(F)** |
| | **M1(T)** | M6(V) | M7(V) | M7(V) | M7(V) | M8(V) | M8(V) | M8(V) | M9(V) | M9(V) |
| | **M2(T)** | M6(V) | M6(V) | M7(V) | M7(V) | M7(V) | M8(V) | M8(V) | M8(V) | M8(V) |
| **Central** | **M3(T)** | M6(V) | M6(V) | M6(V) | M7(V) | M7(V) | M7(V) | M7(V) | M8(V) | M8(V) |
| **Heater** | **M4(T)** | M5(V) | M6(V) | M6(V) | M6(V) | M6(V) | M7(V) | M7(V) | M7(V) | M7(V) |
| **Output** | **M5(T)** | M5(V) | M5(V) | M5(V) | M6(V) | M6(V) | M6(V) | M6(V) | M6(V) | M6(V) |
| **Water** | **M6(T)** | M4(V) | M5(V) | M5(V) | M5(V) | M5(V) | M5(V) | M6(V) | M6(V) | M6(V) |
| **Temperature** | **M7(T)** | M4(V) | M4(V) | M4(V) | M4(V) | M4(V) | M5(V) | M5(V) | M5(V) | M5(V) |
| **(T)** | **M8(T)** | M3(V) | M3(V) | M3(V) | M3(V) | M3(V) | M4(V) | M4(V) | M4(V) | M4(V) |
| | **M9(T)** | M1(V) | M1(V) | M1(V) | M1(V) | M1(V) | M1(V) | M1(V) | M2(V) | M2(V) |

M1 to M9 for each fuzzy variable is the linguistic membership function in-sequence, where, M1 is the least membership function, and M9 is the largest membership function.

### 4.3. Fuzzy Logic Inference Engine (FIE)

"Mamdani" FIE is used with "Minimum" for "And", "Probabilistic" for "Aggregation", and "Centroid" for "Defuzzification".

### 4.4. Simulink Model

The complete Simulink model is illustrated in Figure 6.

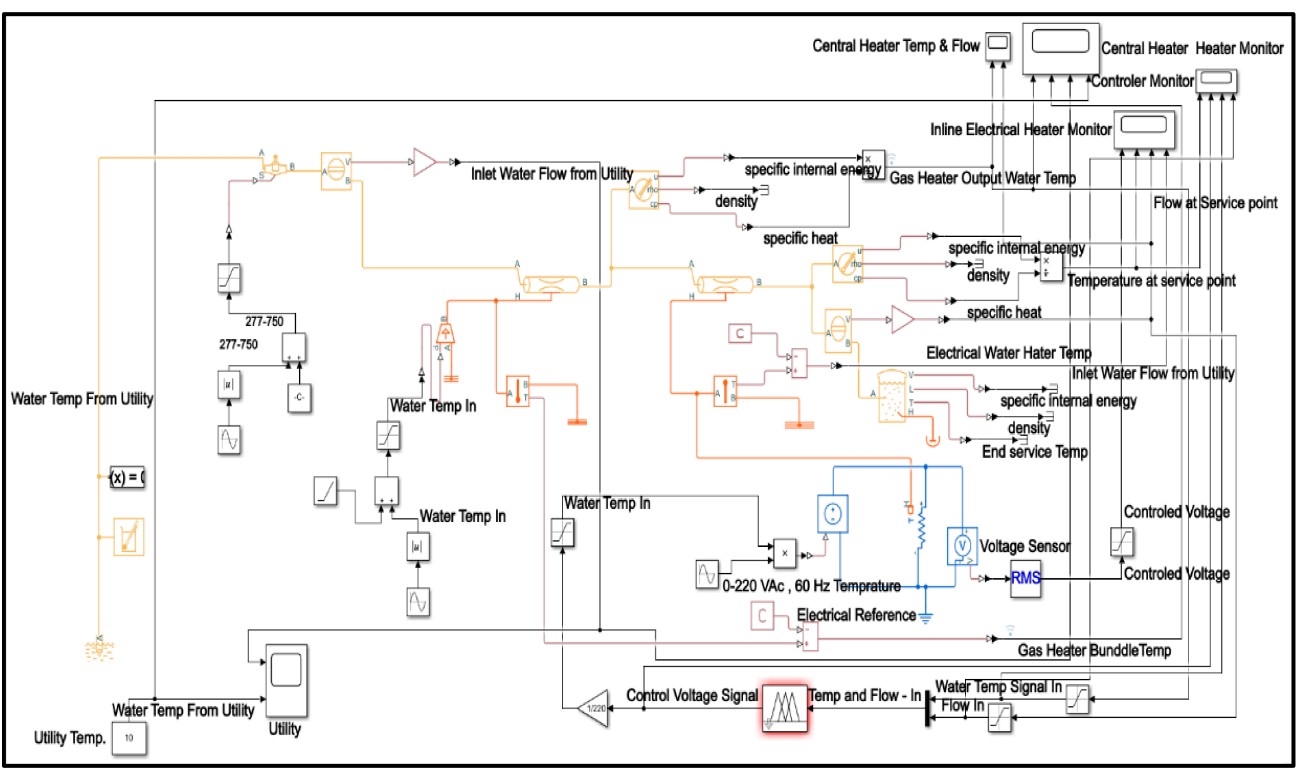

**Figure 6.** Simulink model.

## 5. Testing and Commissioning

In order to test the performance of the proposed system, two sets of validation cases are carried out. "Extreme Cases" and "Normal Operation Cases".

### 5.1. Extreme Test Cases

Two case are carried out with extreme fluctuation in the measured input variables, water flow and water temperature. Extreme limits of the input variables "Temperature" and "Flow" are used to simulate this extreme cases. The purpose of this set of test cases is to test the fuzzy controller under extrema operation condition and observe its performance to determine the proposed system limitations.

#### 5.1.1. Nonlinear "Slow" Flow Rate Fluctuation

The first case is nonlinear "Slow" fluctuation of the flow compared with the nonlinear fluctuation of the water temperature received from the central water heater. Figure 7 illustrates the performance of the fuzzy controller and this extreme test case.

#### 5.1.2. Nonlinear "Fast" Flow Rate Fluctuation

In the second case, nonlinear "Fast" fluctuation of the flow rate compared with the nonlinear fluctuation of the water temperature received from the central water heater is applied. Figure 8 shows the performance of the fuzzy controller and this second extreme test case.

#### 5.1.3. Observations and Discussion on the Extreme Test Cases

Table 2 summarizes important observations from the two extreme cases.

Refereeing to the "Time and Temperature Relationship to Extreme e Burns" provided in [16], In this reference, it is also given that The American Journal of Public Health prefers a maximum temperature of 120 °F (48.89 °C) for hot water. Table 3 summarizes the applicable temperature values to the extreme test cases under discussion.

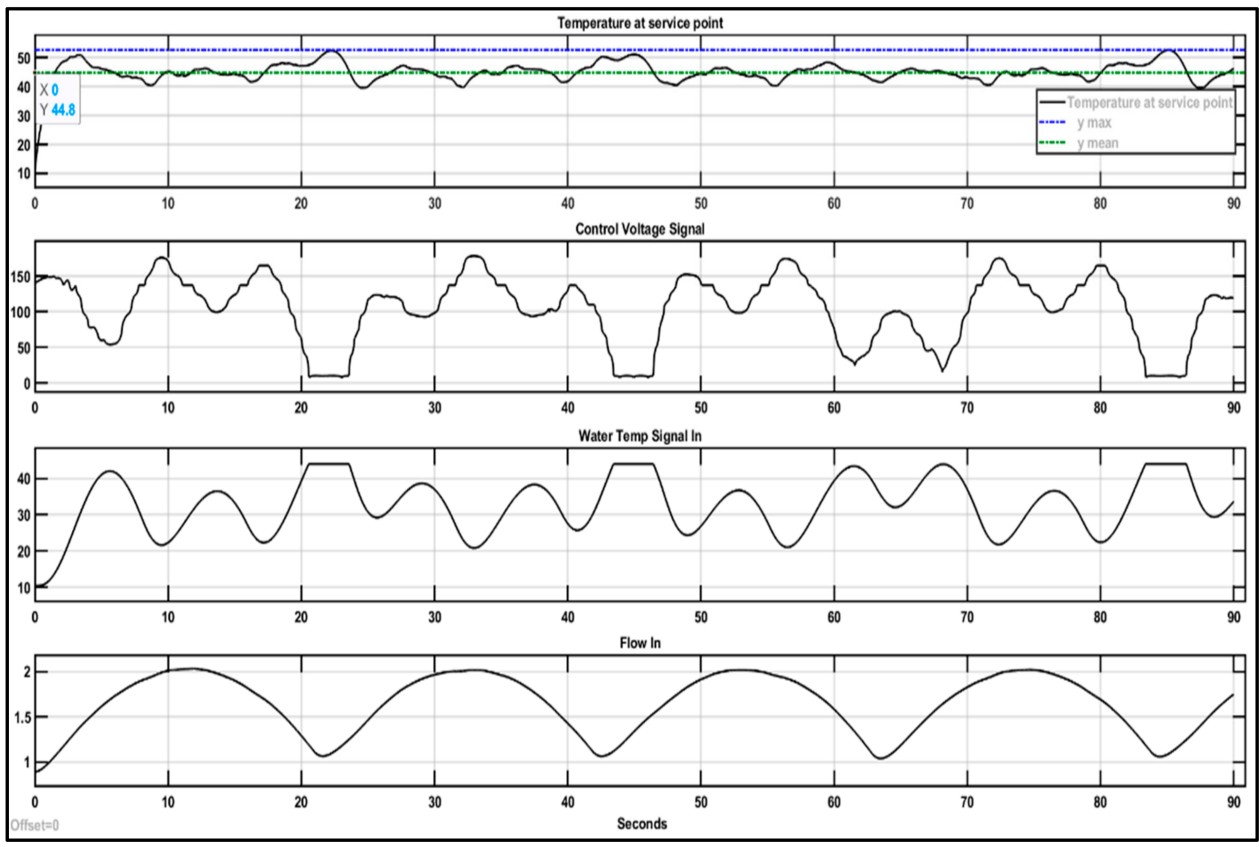

**Figure 7.** Nonlinear "Slow" flow fluctuation performance of the fuzzy controller.

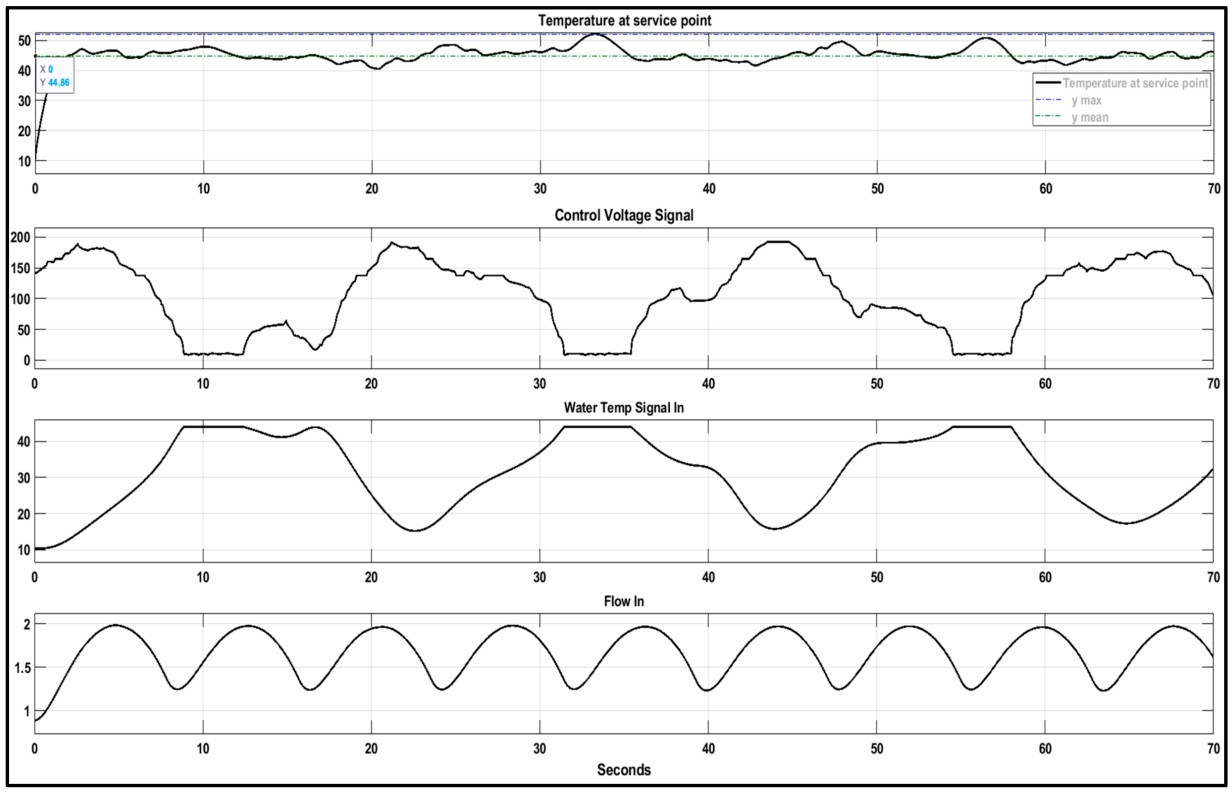

**Figure 8.** Nonlinear "Fast" flow fluctuation performance of the fuzzy controller.

**Table 2.** Extreme test cases observations summary.

|  | Nonlinear "Slow" Flow Fluctuation | Nonlinear "Fast" Flow Fluctuation |
|---|---|---|
| **Maximum Temperature** | 52.62 °C | 52.07 °C |
| **Time to Drop Water Temperature from Maximum (Hot} to Safe Warm (49 °C)** | 1.1 s approx.. | 1.2 s approx. |
| **Minimum Temperature** | 39.61 °C | 39.61 °C |
| **Mean Temperature** | 40.05 °C | 44.86 °C |

**Table 3.** Time and Temperature Relationship to Extreme e Burns.

| Celsius Temperature | 2nd Degree Burn No Irreversible Damage | 3rd Degree Burn Full Thickness Injury |
|---|---|---|
| 45° | 2 h | 3 h |
| 47° | 20 min | 45 min |
| 48° | 15 min | 20 min |
| **\* 49°** | **8 min** | **10 min** |
| 51° | 2 min | 4.2 min |
| 55° | 17 s | 30 s |

\* 48.89 °C.

Comparing the maximum temperature and the time recorded from the extreme cases in Table 2 and compare it with the recommended safe temperature and time in Table 3, we can conclude that the proposed water heater controller maintains the shower water temperature at safe temperature continually. It also can be noticed that in both cases, the water mean temperature is kept at the most satisfactory temperature for shower (44 °C approximately).

*5.2. Normal Operation Test Cases.*

In this set of test cases, three study cases are carried out as typical scenarios while shower is used; fluctuation of water temperature received from central heater at maximum flow rate, linear increase of water temperature received from central heater at maximum flow rate and at minimum flow rate.

5.2.1. Central Heater Temperature Fluctuation at Maximum Flow Rate

In this case, two fluctuation cycles in the central water heater is assumed, in the first cycle, the water temperature after reaches the pre-set value of 44 °C, drops to 15 °C. In the second cycle, the temperature raises again to 44 °C, then drops to 24 °C before it reaches its continue steady stat temperature of 44 °C. The performance of the fuzzy controller is illustrated in Figure 9.

5.2.2. Linear Increase of Central Heater Water Temperature at Maximum Flow

In this case, it is assumed that the cold water with flow rate of 2 gpm in the hot water pipeline needs 3 min to be flushed out till the hot water from central heater arrives to showerhead. Linear rate of increase in central heater water temperature is considered with disturbance in the central heater for short period in the mid time. The performance of the fuzzy controller is illustrated in Figure 10.

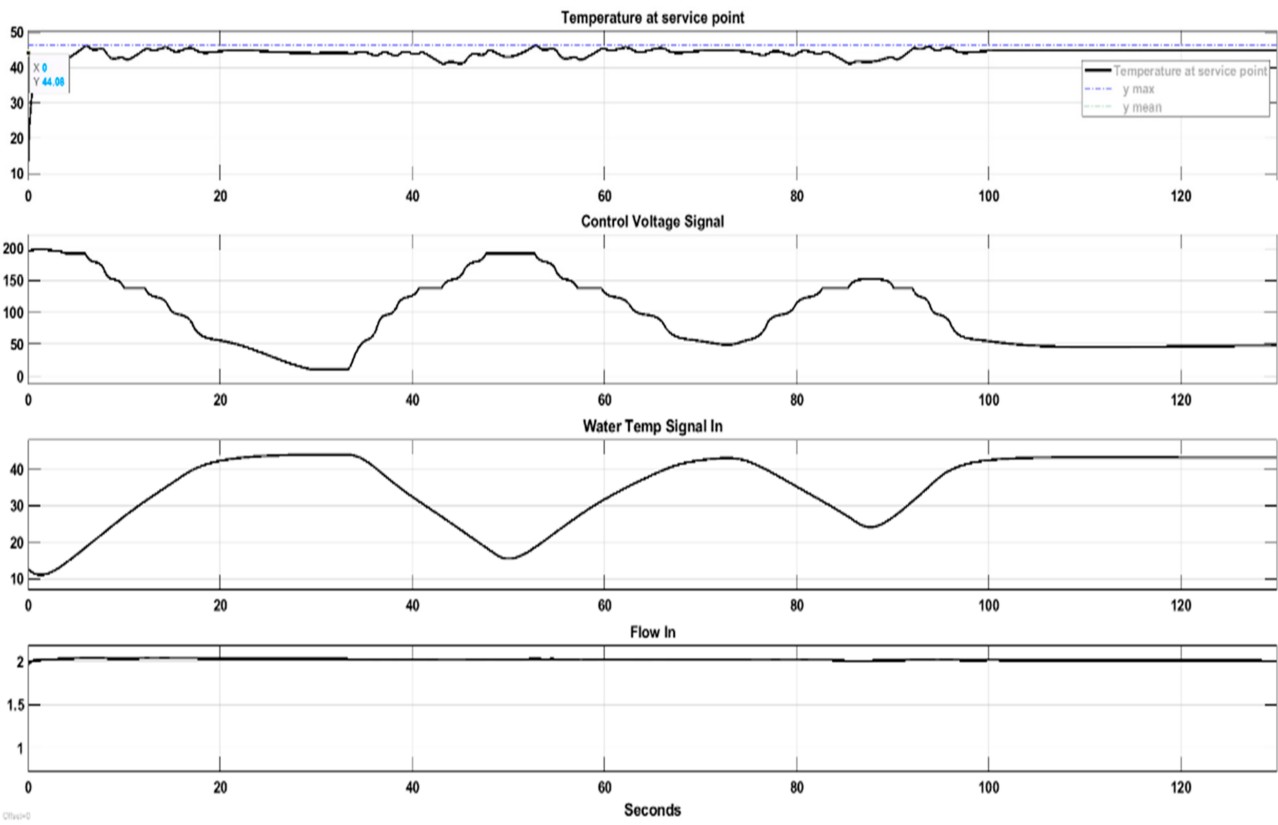

**Figure 9.** Performance of the fuzzy controller with Central Heater Fluctuation at maximum flow.

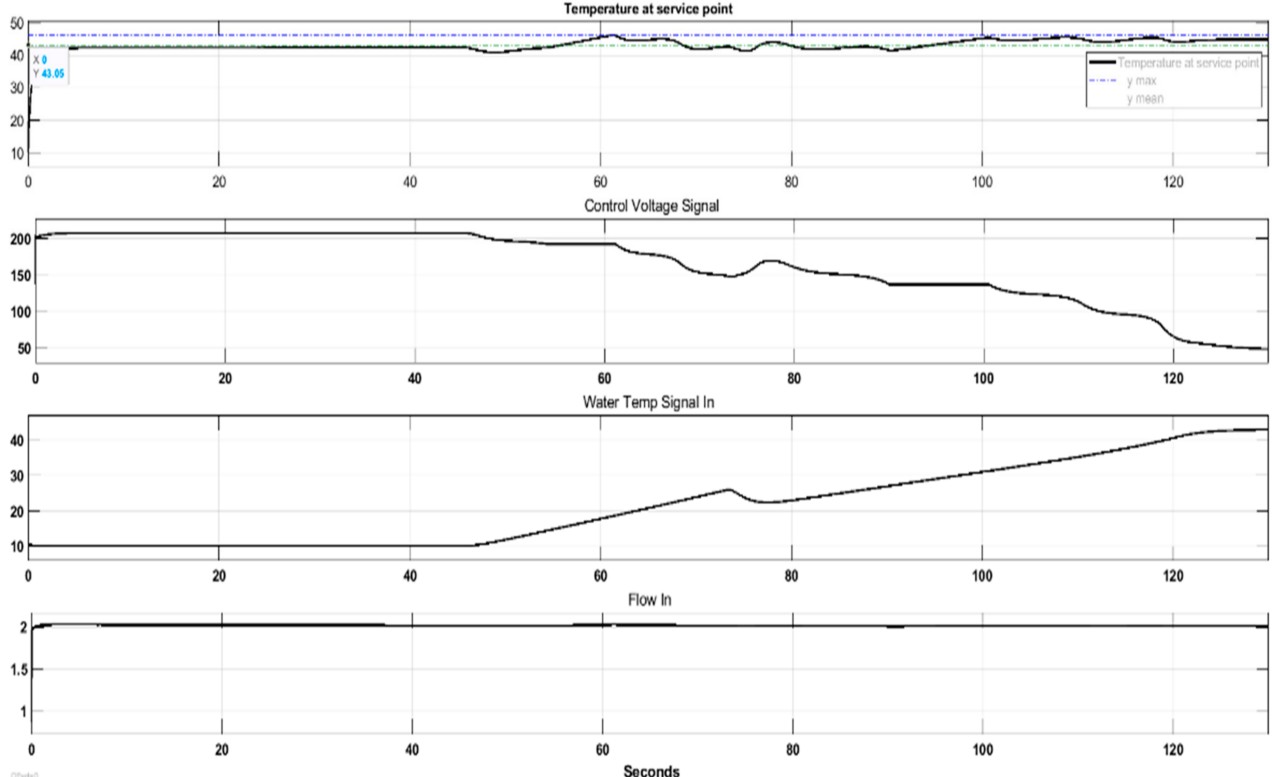

**Figure 10.** Performance of the fuzzy controller with Linear increase of Central Heater water temperature at maximum flow.

### 5.2.3. Observations and Discussion on the Normal Case Test

From the three cases included in the Normal operation tests, Table 4 summaries the water temperature values at head shower.

**Table 4.** Normal operation test cases observations summary.

| In-Line Warm Water | Central Heater Temperature Fluctuation at Maximum Flow | Linear Increase of Central Heater Water Temperature at Maximum Flow | Linear Increase of Central Heater Temperature at Minimum Flow |
|---|---|---|---|
| **Output Maximum Temperature** | 46.32 °C | 46.23 °C | 50.04 °C |
| **Time to drop water temperature from maximum (Warm) to safe Warm (49 °C)** | 0 | 0 | 1.5 s approx. |
| **Minimum Temperature** | 41.04 °C | 40.96 °C | 44.44 °C |
| **Mean Temperature** | 44.08 °C | 43.05 °C | 45.85 °C |

The mean temperature that the user will feel during the operation of the inline electrical heater until the water of the central heater reach steady state temperature of 44 °C is from 43.05 °C to 45.85 °C. In case the users change the water flow rate between minimum to maximum, the average temperature during the change of the flow rate is 44.08 °C. These results for controlled temperature values are very satisfactory result as the rate that skin temperature changes influences how readily people can detect the change in temperature. If the temperature changes very slowly, for example at a rate of less than 0.5 °C per minute, then a person can be unaware of a 4–5 °C change in temperature, provided that the temperature of the skin remains within the neutral thermal region of 30–36 °C. If the temperature changes more rapidly, such as at 0.1 °C/s, then small decreases and increases in skin temperature are detected [17,18].

## 6. Economic Analysis

In this section, techno-economic analysis includes direct and indirect benefit that can be obtained by the installation of the inline heater equipped with proposed fuzzy controller to solve the domestic problem of waiting hot water for the shower use, which result wasting fresh water [19–24].

### 6.1. Direct Benefits

In this part, following factors are considered to calculate the direct benefit:

(a) The initial cost of installing inline heater with fuzzy controller.
(b) Water saving.
(c) Operation & maintenance (O&M) cost.

### 6.2. Indirect Benefits:

Three additional factors are considered as Indirect Benefit can be also obtained by installing this proposed inline heater:

(a) Increase water sales opportunity.
(b) Pollution Cost.
(c) Electrical Energy Saving.

For Direct Benefits, calculation for initial of installation cost, water supply cost, wastewater tariff and O&M are collected based on Azerbaijan market 2021. The estimated number of the inline electrical heater unites is calculated based on the assumption of one unit shall

be installed in each household. The total number of household in Azerbaijan as per the latest data 2020 is 2,170,100 units [24,25].

In Direct Benefits, calculation for initial installation cost, water supply cost, wastewater tariff and O&M are collected based on Azerbaijan market 2021. The estimated number of the inline electrical heater unites is calculated based on the assumption of one unit shall be installed for each household. The total number of households in Azerbaijan as per 2020 is 2,170,100 [25–27].

Also, greenhouse gas release occurs when water is extracted and transported to industrial, agricultural, and residential areas for use, which consumes energy and materials. For water extracting from different sources and its transporting and processing, the carbon footprints can be very different [28]. In pollution cost calculation under Indirect Benefits in this paper, rate of greenhouse gas emission is considered based on the water availability mostly from natural surface and underground resources. Minor amounts produced from distillations plants are ignored in this calculation [29]. The Emission Factors that considered for water supply and wastewater treatment are 0.344 kg–0.708 kg $CO_2$ m$^{-3}$ respectively [30]. Based on these factors, the associated greenhouse gas emissions in "tons" of carbon dioxide is calculate by the following equation:

$$\text{Greenhouse gas emissions} = \text{Data} \times \text{Emission Factor} \tag{18}$$

Accordingly, due to the reduction of the $CO_2$ emission, and hence less pollution Carbon credit can be calculated. Carbon credits based on current market is 6 euro/ton. Where, $CO_2$ emission is considered to be in kg kW$^{-1}$ h$^{-1}$. Considering typical energy demand to treat, transfer and supply one m$^3$ from surface water reservoir is 0.43 kWh m$^{-3}$ [31], and assuming Euro to USD exchange rate of 1.2, the annual saving in pollution reduction can be calculated as following:

$$\text{Annual Saving in Polution} = \frac{\text{kg}}{\text{kWWhr}} \times \frac{\Delta\text{kWhr}}{1000} \times 6 \times 1.2 \ \$ \tag{19}$$

As mentioned above, to pump the water from surface water reservoir, transport the water and treat the water to be suitable for domestic use, electrical energy must be spent with rate of 0.43 kWh/m$^3$. Therefore, in case an amount of water shall be saved, the respective electricity shall be saved as well. Direct and indirect benefits' calculation are summarized in Table 5.

**Table 5.** Summary of direct and indirect benefits' techno-economy calculation.

| | | Directs Benefits Calculation | |
|---|---|---|---|
| **S/N** | **Description** | **Investment of Installation 3 kW Inline Electrical Heater with Controller** | **Remarks** |
| 1 | Initial Cost Per Unit | $32 | [32] |
| 2 | Total Quantity | 2,170,100 | |
| 3 | Total Quantity Cost | $69,443,200 | |
| 4 | Installation Cost Per Unit | $15 | Fittings and labor work. |
| 5 | Total Installation Cost | 32,551,500 | |
| 6 | Minimum Lifetime of Inline Heater (Years) | 5 | |
| a | **Therefore Initial Investment for 5 Years** | **$101,994,700** | Minimum 5 years operation |

**Table 5.** *Cont.*

| S/N | Description | Saving of Wasted Water | |
|---|---|---|---|
| colspan Directs Benefits Calculation |
| 1 | Wasted Time Per Event | 3 min | |
| 2 | Wasted Water Per Event | 6 Gallon | Maximum Flow Rate = 2 gpm |
| 3 | Number of Events Per Day | 10,000,000 | Azerbaijan population 10,067,100 [25] |
| 4 | Total Water Saved in m$^3$ Per Day | 272,765.71 | |
| 5 | Annual Water Saved M$^3$ | 74,669,612.54 | Assuming operating time is 9 months |
| 6 | Residential Water Supply Fees \$/M$^3$ | 0.587544066 | The tariffs include wastewater discharge and VAT. [27] |
| 7 | Annual Water Saving Cost | 43,871,687.76 | |
| b | **Therefore Total Water Saving Cost in 5 years** | **219,358,438.8** | **Minimum 5 years operation** |

| S/N | Description | O&M | Remarks |
|---|---|---|---|
| | Service Life Range | 5 Years | |
| 1 | Operating Time Per Event | 3 min | |
| 2 | Average Consumption Power Per Event | 2 kW | |
| 3 | Number Of Events Per Day | 10,000,000 | |
| 4 | Annual Operating Consumption | 273,750,000 | |
| 5 | Azerbaijan Electricity Prices/kWh | 0.041 | 0.07 Manat/kWh |
| 6 | Annual Operating Cost | 11,223,750 | |
| 7 | Maintenance Cost | 0 | Negligible in 5 years as no moving parts. |
| c | **Therefore Savings Total O&M Cost for 5 Years** | **\$56,118,750** | |

| S/N | Description | Increase Water Sales opportunity | Remarks |
|---|---|---|---|
| colspan Indirect Benefits Calculation |
| 1 | Annual amount of fresh water sale opportunity | 74,669,612.54 | |
| 2 | Fees for Consumers, which use the water as an row material \$/m$^3$ | 7.3 | 12.3manat The tariffs include wastewater discharge and VAT. [27] |
| **3** | **Annual Sales opportunity \$** | **545,088,171.52** | |
| d | **Water Sales opportunity in 5 years** | **2,725,437,781 \$** | |

| S/N | Description | Utility Electricity saving | Remarks |
|---|---|---|---|
| 1 | Annual Saved water in 1 years | 74,669,612.54 | |
| 2 | kWkr required to produce 1 m$^3$ fresh water | 0.43 | |
| 3 | Annual Energy saved kWh | 32,107,933.39 | |

**Table 5.** *Cont.*

| | Indirect Benefits Calculation | | |
|---|---|---|---|
| S/N | Description | Utility Electricity saving | Remarks |
| 4 | Minimum electricity fees need to produce I m$^3$—\$/kWh | 0.041 | 0.07 Manat/kWh |
| 5 | Annual electricity cost saving | 1,316,425.269 | |
| **e** | **Electricity cost saving for 5 year** | **6,582,126.345** | |
| S/N | Description | Pollution cost | Remarks |
| 1 | Saved water in 5 years | 74,669,528.25 | |
| 2 | Emission Factors for water supply $CO_2$ kg /m$^3$ | 0.344 | |
| 3 | Emission Factors for wastewater treatment $CO_2$ kg/m$^3$ | 0.708 | |
| 4 | Annual saving in pollution reduction | 565,577.5132 | |
| **f** | **Pollution cost in 5 years** | **2,827,887.566** | |

Bases in the techno-economic calculations illustrated in Table 5, two successful, profitable and household satisfactory scenarios can be determined and summarized. The first scenario is based on Water Utility to provide the required investment to install one inline water heater in each residence. The second scenario is based on that each household to install at least one inline water heater in their residence. In Table 6, the analysis focuses more on the "Indirect Benefit" and highlights the National (Government) economic benefit that can be achieved from the investment in this application.

**Table 6.** Techno-economy analysis for "Indirect Benefit".

| | Net Cost Saving Using Sales Opportunity (Investment by Water Utility) | Net Cost Saving without Using Sales Opportunity (Investment by Households) | Remarks |
|---|---|---|---|
| Applicable S/N | a, b, c, d | c, e, f | |
| Saving in 5 Year | −101,994,700 − 219,358,191 + 56,118,750 + 2,725,437,781 = \$2,460,203,392 | 56,118,750 + 8,508,602.349 + 2,827,887.566 = \$67,455,239.92 | Investment shall be done by Utility In case the water shall be sold, the pollution amount shall not be changed |
| Annual Saving | **\$492,040,678.44** | **\$13,491,047.98** | |
| Payback Period in Years | **0.207 Year (2.5 months)** | **0 Year** | |
| Annual "ROI" in Percentage * | **382.42%** | **Only profit** | |

* ROI: Annual Return on Investment.

Where:

$$\text{ROI\%} = \frac{\text{Gain from Investment} - \text{Cost Investment}}{\text{Cost Investment}}\% \qquad (20)$$

The second analysis focuses on the "Direct Benefits" that the householders can achieve, if the investment is done by either water utility/government (First Scenario) or by households (Second Scenario). Table 7 illustrate this "Direct Benefits".

Obviously, there is another scenario, which is ignored intentionally, as it is a result of the first two scenarios. This scenario is households to provide the investment, and water/government to use the increase water sales opportunity. This scenario is obviously more profitable for the water utility.

**Table 7.** Techno-economy analysis for "Direct Benefit".

| | Before Installation of Inline Heater | After Installation of Inline | | Remarks |
| --- | --- | --- | --- | --- |
| | | Investment by Water Utility | Investment by Households | |
| **Time Saving Per Person Per Year** | - | 821.25 min, 13.65 h | 821.25 min, 13.65 h | Assuming 3 min wasting per shower event. |
| Wasted Water Annual Cost | 4.4 $ (7.5 Manat) | 0 | 0 | |
| Annual Cost in Saving Wasted Water | 0 | (0.9 $) 1.5 Manat | 1.57 $ (2.66 Manat) | |
| **Net Saving In 5 Years** | 0 | (17.6 $) 30 Manat | 14.15 $ (24.1 Manat) | |

Useful indices can be calculated from the Techno-economy analysis to determine the saving amount that can be achieved for the installation of one inline heater:

$$\text{Savng index} = \frac{\text{Annual Total Net Saving}}{\text{Number of installed inline heater}} \qquad (21)$$

From the result in Tables 5 and 6 and Equation (20), following indices can be determined:

- Annual Saving index considering Sales opportunity: 226.74 $/inline heater unit
- Annual Saving index without considering Sales opportunity: 6.2 $/inline heater unit.

## 7. Conclusions

This paper discussed important domestic problem related to waiting hot water for the shower use till reach satisfactory temperature, which result a lot of wastage in fresh water, Comprehensive survey is carried out to evaluate the available solutions, if any, to this problem. Some practical trials were carried out by different researchers, but the results were not satisfactory to the users, either because of the high cost or because of the results of the solution was ineffective. In this paper, a new concept of using in-line electrical heater with fuzzy MISO logic controller is introduced. The function of the fuzzy controller for the inline electrical heater that supposed to be installed in suitable location before the showerhead is to measure the water temperature and its flow rate before the showerhead, as control input-variables, and decide the operating voltage of the heater, as control output-variable. Simulink model is built and tested for this solution. Two types of tests were implemented, "Extreme test cases" and "Normal Operation test cases". The results prove high accuracy and safe operation of the proposed system.

Techno-economic study was carried out to validate the proposed solution from economic point of view. The study included "Direct Benefits" and "Indirect Benefits" that can be achieved by the implementation of the proposed system, Azerbaijan was taken as example in this techno-economic study. Two successful, profitable for water utility and satisfactory for users, scenarios were determined and summarized. The first scenario considers the investment on supply one inline water heater for each household is done by water utility. The second scenario considers the investment is done by households.

For Indirect Benefit (mostly National Benefit), the results showed high economic benefits. Also, important environmental benefits are achieved from the reduction of the greenhouse gas emission. Annual increase in fresh water sales opportunity of 545,088,171.52 $ ($2,725,437,781 in 5 years), with very short payback period for the amount of the investment (2.5 months) and very high annual "ROI" (382.42%) are expected, provided that the proposed system is installed in all households (2,170,100 Units). If fresh water sales opportunity is not considered, and the investment is done by households, the proposed solution was found to be still attractive with annual profit of $13,491,047.98 for the water utility.

For users' Direct Benefits, the annual utility bills reduced by approximately 3.6 $ (6Manat) in first scenario, or approximately 2.8 $ in second scenario. However, the main benefit is the satisfaction of the households as will not be further time waste in waiting hot water for shower, this can save approximately half day per year for each person.

Important indices were calculated to determine saving amount of money that can be achieved for the installation of one inline heater. These indices are useful for decision makers that will provide the required investment for such important national project.

**Funding:** This research received no external funding.

**Institutional Review Board Statement:** Not applicable.

**Informed Consent Statement:** Not applicable.

**Data Availability Statement:** Data available in a publicly accessible repository. The required information were listed in "Reference" section for all data used in the paper.

**Conflicts of Interest:** The author declares no conflict of interest.

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
