# Peer review of "Sustainable Economic Solution to Prevent the Wasted Cold Water in the Start of Shower Time Using Low Cost In-Line Electrical Heater with MISO Fuzzy Logic Controller"

_water, doi:10.3390/w13111552_

Round 1

Reviewer 1 Report

Muhammad M. A. S. Mahmoud presented a study to optimize a hot water system for the shower using in-line electrical heater with fuzzy logic controller.  The results are very interesting, in especial the tecno-economic study that evaluated different scenarios of this system.

The authors need to double check the manuscript in terms of format and grammatical errors in most of the sections. There are several typos, variables from different equations that were not explained, incorrect symbols, missing information about the program used and the quality of the images should be improved before publication. It is recommend that the author write in third person and avoid the use of you, they, and we trough the manuscript.

Format comments

Line 17-19: Please Add a sentence with the relevant results (quantitative if it is possible) to the abstract. For instance info from the tecno-economic study.

Line 45: “…and flow rate aren’t quite…” Please Change aren’t by are not.

Line 48: “… When you’re …”Please change you’re by the user is.

Line 52: “..If you …” it is recommended that the author write in third person. The author can use “the user” instead of “you”. Same comment for Line 58: we’re.

Lines 80 and 82: It is important that the author define all of the variables used in the graphs. For instance, there is not an explanation about S, BSN, SHR, W H.

Line 102: Caption of the figure 3 is incomplete. There is no information about the significance of the color of each line and the quality of the image is not good enough.

Lines 104-107: “This method ….hot water pipe” The sentence is too long. It is recommend to break the sentence in two or three short sentences.

Line 211: There is a typo inside of the image. Please change worm by warm.

Line 228: Variable “R” was not defined.

Line 229: Please change R0 by Ro.

Line 235: “i current”. Please change the previous by “I: current”.

Line 250: use the same symbol that you use in your equation 4 to define the density.

Line 257, 258, 267, 268 and 270, 283, 288, 322: The variables L, moutlet, minlet, D, V, u and D again were not defined.

Line 273: There is no “N” variable in the equations 7, 8 , 9 , 10

Line 297: There is no “ Qconv” in your equation 13

Line 307: “0.5x(m….)” If the “x” in the previous expression represents a product is better to use other symbol to avoid confusions.

Line 311: Please change “NU” by “Nu

Line 329: Please change “40.5°C 43.5°C” by “40.5°C to 43.5°C”

Line 334: Please change “10°C, 44°C” “in the range from 10°C to 44°C”

Line 340: Please change “section II” by “section 2”

Line 389: Please change the format of your table 2, because is not clear the header of each respective column

Line 426, 428 Please check the symbol of Celsius degress. The real symbol should be °C instead of â—¦C

Line 471, 472, 475, 477; Please change CO2 by CO2

Line 483: please change m3 by m3

Line 430: please double check if the correct value is 44.408°C or 44.08°C

Questions

1. On line 144, the author showed in the “figure 4” a comparison in terms of delivery time for the different plumbing system. Does the author obtain these results through different experiments? If so the authors should move this figure to the results section and provide the respective clarification in the manuscript.

2. The author mention on lines 186-187 that the minimum temperature water is 10°C. Is this a specific value for the Republic of Azerbaijan or a general value for several places? Add the respective clarification in the main manuscript.

3. The author mention on line 341 that Simulink was used to run the different simulation but details about the program were not provided in the methodology section. Please add the respective information including for instance version, capabilities, etc.

4. The author showed the simulation model on line 344 but there is not additional information or details about the model. Please add the respective information.

5. Line 442. Please add additional references to this section to avoid the inappropriate use of self-citations by the author.

6. Can the system proposed by the author be extended to the kitchen sink? If so, how complex would be this addition to your design?

Author Response

Comments and Suggestions for Authors

Format comments

  1. Line 17-19: Please Add a sentence with the relevant results (quantitative if it is possible) to the abstract. For instance info from the tecno-economic study.

A statement is added in line 18: “ ,that can increase fresh water sales opportunity to 545,088,171.52 $,”

  1. Line 45: “…and flow rate aren’tquite…” Please Change aren’t by are not.

Done: Line 47 reads:  “The effects of pipe diameter and flow rate are not so intuitive”

  1. Line 48: “… When you’re …”Please change you’re by the user is.

Done : Line 49 reads “…. .When the user is waiting for hot wat-“

  1. Line 52: “..If you…” it is recommended that the author write in third person. The author can use “the user” instead of “you”. Same comment for Line 58: we’re.

Done: Line 53 reads: “…. If the user has

          Line 59 read: ” … larger houses are being built.”

  1. Lines 80 and 82: It is important that the author define all of the variables used in the graphs. For instance, there is not an explanation about S, BSN, SHR, W H.

Done: Line 94 -95: Explanation is given.

                            Figure Quality is improved

  1. Line 102: Caption of the figure 3 is incomplete. There is no information about the significance of the color of each line and the quality of the image is not good enough.

Done: Line 115 -117: Explanation is given.

Figure Quality is improved

  1. Lines 104-107: “This method ….hot water pipe” The sentence is too long. It is recommend to break the sentence in two or three short sentences.

Done: Line 121: The sentences splitted into in two shorter sentences.

  1. Line 211: There is a typo inside of the image. Please change worm by warm.

Done:  Figure 5 correctd

  1. Line 228: Variable “R” was not defined.

Done: Line 249: “R: Temperature-dependent resistor”

  1. Line 229: Please change R0 by R

Done: Line 247 R0 changed by Ro

  1. Line 235: “i current”. Please change the previous by “I: current”.

Done: Line 245 and 253 read “I”

  1. Line 250: use the same symbol that you use in your equation 4 to define the density.

Done: Line 275 reads: r

  1. Line 257, 258, 267, 268 and 270, 283, 288, 322: The variables L, moutlet, minlet, D, V, u and D again were not defined.

Done: Variables’ Definition of Equation (4-10) are checked and revised (Repetition is avoided)

  1. Line 273: There is no “N” variable in the equations 7, 8 , 9 , 10

Done: Spilling mistake corrected in Equation 7 and 9 that are function of the Kinematic Viscosity N

  1. Line 297: There is no “ Qconv” in your equation 13

Done: Spilling suffix mistake in equation 13 is corrected.

  1. Line 307: “0.5x(m….)” If the “x” in the previous expression represents a product is better to use other symbol to avoid confusions.

Done: The confusion is avoided by using : mavg = (minlet – moutlet)/2

  1. Line 311: Please change “NU” by “Nu

Done: Please see point 14.

  1. Line 329: Please change “40.5°C 43.5°C” by “40.5°C to 43.5°C”

Done: Line 350 reads: “40.5°C to 43.5°C”

  1. Line 334: Please change “10°C, 44°C” “in the range from 10°C to 44°C”

Done: Line 355 reads: “10°C to 44°C”

  1. Line 340: Please change “section II” by “section 2”

Done: Line 360 Reads: “section 2”

  1. Line 389: Please change the format of your table 2, because is not clear the header of each respective column

Done: Header format of Table 2 is revised.

  1. Line 426, 428 Please check the symbol of Celsius degress. The real symbol should be °Cinstead of â—¦C

Done, Symbols corrected in Table 2 and Table 4

  1. Line 471, 472, 475, 477; Please change CO2 by CO2

Done: CO2  is used in all paper.

Line 483: please change m3 by m

Done; Line 504 reads: m3

Line 430: please double check if the correct value is 44.408°C or 44.08°C

Done: Typing mistake is corrected. Line 450 reads: 44.08°C

Questions

  1. On line 144, the author showed in the “figure 4” a comparison in terms of delivery time for the different plumbing system. Does the author obtain these results through different experiments? If so the authors should move this figure to the results section and provide the respective clarification in the manuscript.

This figure is part of the survey and the reference is given in the [9]. Therefore, I believe this part is written in the correct section. 

  1. The author mention on lines 186-187 that the minimum temperature water is 10°C. Is this a specific value for the Republic of Azerbaijan or a general value for several places? Add the respective clarification in the main manuscript.

As per Engineering standard, the recommended temperature for pump stations is between 50 °F (10°C) and 90 °F (32°C) to prevent equipment and process lines from freezing or overheating. Heat tracing system is usually used to supplement ambient heating in cold weather. Based on this standard, also water heaters are designed and sized considering the inlet water temperature is 10oC.     

  1. The author mention on line 341 that Simulink was used to run the different simulation but details about the program were not provided in the methodology section. Please add the respective information including for instance version, capabilities, etc.

Simulink is a well know tool in the famous software “MATLAB”, I have already added it the reference list [15]. As the paper is not related mainly to software subject, I can see that referring to Simulink should be enough, and any reader is interested to know more about Simulink, she/he can refer to the reference. I added the version that I have used in the reference [15].

  1. The author showed the simulation model on line 344 but there is not additional information or details about the model. Please add the respective information.

In this section I amusing standard and common membership function that used in Fuzzy logic, therefore, I believe it is not required to elaborate more, but as long as it is a comment from reviewer, I will add some more information. Line 369-371.

  1. Line 442. Please add additional references to this section to avoid the inappropriate use of self-citations by the author.

Line 487: Additional references is added, [25-27].   

  1. Can the system proposed by the author be extended to the kitchen sink? If so, how complex would be this addition to your design?

Yes, the same system with the same controller exactly can be used in kitchen sink application, but usually the fresh water is wasted considerably   in shower as the full body shall be under the water, but in kitchen sink, hands in general is not so sensitive like all body under water. Also in kitchen sink, the water in the beginning can be poured directly on the dishes until warm water reaches the tap, and in this case the water is not actual wasted. So, I cannot see that this is a real problem. However, for more luxurious kitchen, same system can be used.

 Please find the revised version after in cooperating all reviewer’s comments. 

Reviewer 2 Report

This paper discusses the domestic problem of waiting for hot water for shower use till it 7 reaches a satisfactory temperature, which results in a lot of wastage in fresh water. The paper study is innovative and the content is highly significant.

The presentation quality can be improved to make sure that all the important elements are not left out and the article is flowing. the flow is interrupted by too many short subsections that 

Page 2 (65-67): please introduce the different approaches followed to solve the problem of waiting for hot water and the consequent wasting of cold water in the paragraph first before explaining them in their respective sub-sections. 

154: In [11], project seeks to - please correct the referencing

Recheck the assumptions made in designing Simulink Model for "Normal Operation Cases" so that you ensure that the direct and indirect benefits are not unrealistic. Is there a way to make the words in Figure 6: Simulink Model, legible? The same applies to 4, 5, 7 - 9.

Discuss the research limitations. For instance, there is an assumption that cold water is often wasted. In some houses, the cold water is collected and used for other purposes eg. if it is the kitchen sink it is used to soak and wash dishes.

Author Response

Comments and Suggestions for Authors

  1. This paper discusses the domestic problem of waiting for hot water for shower use till it reaches a satisfactory temperature, which results in a lot of wastage in fresh water. The paper study is innovative and the content is highly significant.

Thank you for the Reviewer Positive evaluation

  1. The presentation quality can be improved to make sure that all the important elements are not left out and the article is flowing. the flow is interrupted by too many short subsections that

Page 2 (65-67): please introduce the different approaches followed to solve the problem of waiting for hot water and the consequent wasting of cold water in the paragraph first before explaining them in their respective sub-sections.

Thank you for the comment.

Kindly not that rom line 36 to 64, the paper discusses in detail the problem and the factors affect the problem, and then, from Line 65 to 181, the previous three main efforts/design that were done to solve this problem are discussed in details. Then from line 182 the paper proposes the new solution.        

  1. 154: In [11], project seeks to - please correct the referencing

Web site and DOL are added.

  1. Recheck the assumptions made in designing Simulink Model for "Normal Operation Cases" so that you ensure that the direct and indirect benefits are not unrealistic. Is there a way to make the words in Figure 6: Simulink Model, legible? The same applies to 4, 5, 7 - 9.

Done, Figures resolution is improved. I checked printed vision, it is clear now.

  1. Discuss the research limitations. For instance, there is an assumption that cold water is often wasted. In some houses, the cold water is collected and used for other purposes eg. if it is the kitchen sink it is used to soak and wash dishes.

This application concentrate mainly on the fresh water that is wasted during the shower waiting for worm water to reach the tap,

The same system with the same controller exactly can be used in kitchen sink application, but usually the fresh water is wasted considerably in shower as the full body shall be under the water, but in kitchen sink, hands in general is not so sensitive like all body under water. Also in kitchen sink, the water in the beginning can be poured directly on the dishes until warm water reaches the tap, and in this case the water is not actual wasted. So, I cannot see that this is a real problem. However, for more luxurious kitchen, same system can be used.  

Please find the revised version after in cooperating all reviewer’s comments.  

Round 2

Reviewer 1 Report

No comments. The author has been significantly improved  the quality of the manuscript and now it warrants publication in Water.